# VoxSet: Sparse Voxel Set Tokenizer for 3D Shape Generation

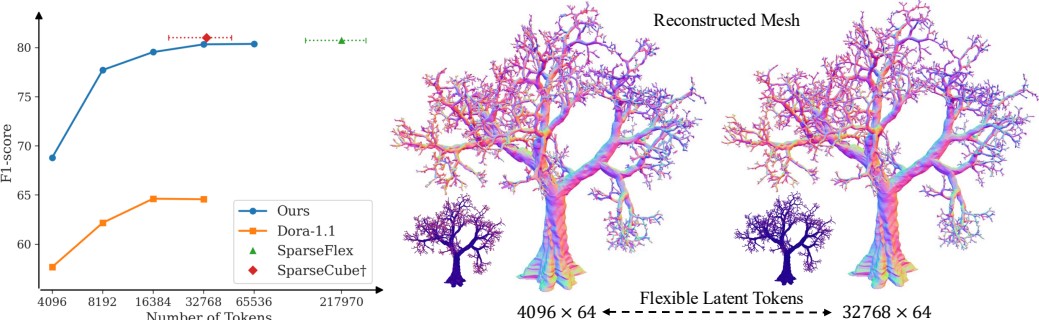

Figure 1: Our VoxSet tokenizer has a compact and flexible latent space while achieving competitive reconstruction quality. For sparse voxel methods, we present the mean and variance of the number of latent tokens in our testing dataset, as they are of variable length. [†]: our implementation of sparse cube tokenizer (Li et al., 2025b; Chen et al., 2025).

## Abstract

3D tokenizers are crucial in latent 3D generative models. Recent sparse voxel tokenizers can reconstruct detailed shapes but produce variable-length latent tokens which necessitate a two-stage generation pipeline. Conversely, vector set tokenizers have fixed-length latent tokens with higher compression but struggle with reconstruction quality. In this work, we introduce VoxSet, a novel tokenizer that combines the strengths of both approaches. Our method employs sparse voxels in the outer layers to capture fine surface details and a vector set bottleneck for high compression. This design achieves high-quality reconstructions while maintaining a compact and fixed-length latent code for different objects, eliminating the extra generation stage required by sparse voxel methods. Experiments demonstrate that VoxSet achieves competitive reconstruction quality compared to sparse voxel tokenizers, while sharing the simpler training and inference pipeline of vector set-based 3D generation models.

## 1 Introduction

High-quality 3D shape generation is important for various applications, such as gaming, AR/VR content creation, industrial design, and robotics. Recent advancements in 3D latent score-based generative models have shown impressive capabilities. These models can generate detailed and diverse shapes by modeling complex data distributions within a compressed latent space (Zhang et al., 2024; Li et al., 2025a; Zhao et al., 2025). A key component of these models is the 3D tokenizer (Zhang et al., 2025; Chen et al., 2024a; Li et al., 2025b; Zhang et al., 2023). It maps high-dimensional 3D shape data, typically signed distance fields (SDF), into a compact latent representation suitable for the generative model. The tokenizer is crucial for both reconstruction fidelity and the efficiency of training and inference in latent generative models. An ideal tokenizer would preserve fine geometric details while maintaining a compact latent space.

Two main tokenizer designs dominate current research: vector set tokenizers and sparse voxel tokenizers. Vector set methods (Zhang et al., 2023; 2024; Li et al., 2025a; Zhao et al., 2025; Wu et al., 2024) represent shapes as unordered sets of latent vectors. These methods achieve high compression and fixed-length latents, enabling simple training of the latent generative models. However, they often

struggle to reconstruct fine spatial details compared to voxel-based approaches. Recently, sparse voxel methods (Xiang et al., 2024; Li et al., 2025b; He et al., 2025; Chen et al., 2025; Wu et al., 2025) have been used to encode 3D geometry into structured latent codes (SLAT) within sparse voxels. These methods achieve impressive reconstruction fidelity, particularly at high resolutions. However, they produce variable-length latents depending on the 3D shape. This variability necessitates a two-stage generation pipeline: the first stage generates the sparse structure, and the second stage generates latent features on this sparse structure. This approach increases complexity in both training and inference and is prone to error accumulation from the first stage model. In summary, no current tokenizer simultaneously offers high-fidelity reconstruction, compact fixed-length representation, and a simple single-stage generation pipeline.

We observe that sparse voxels excel in representing local high-resolution details, while vector sets are more effective at capturing global structure with high compression ratio. This duality motivates our core insight: instead of choosing between them, we can integrate their strengths within a single architecture to leverage the advantages of both. We propose **VoxSet**, a novel tokenizer that applies sparse voxels in the outer layers to capture detailed geometry and compresses it into a fixed-length vector set latent in the middle layers. This hybrid design enables high-resolution reconstruction capabilities on par with sparse voxel methods, while also achieving the compression efficiency of vector set methods. Beyond the performance benefits, **VoxSet** also offers the training and inference advantages of both methods. Compared to vector set methods, our tokenizer unifies the input and output formats as sparse voxels, thereby simplifying tokenizer training. Compared to sparse voxel methods, our fixed-size latent code eliminates the need for the sparse structure stage in the generation process, thus simplifying the overall pipeline.

In summary, our contributions are:

1. We introduce VoxSet, a novel voxel set tokenizer that combines sparse voxel representation with fixed-length vector set compression, achieving high-fidelity reconstruction within a compact latent code.

2. We implement an efficient algorithm to accelerate sparse voxel extraction from meshes, along with sparse marching cubes algorithms to convert sparse voxels back to meshes.

3. Extensive experiments demonstrate that VoxSet significantly enhances reconstruction quality compared to vector set methods when using the same number of latent tokens, and also remains competitive with sparse voxel tokenizers.

## 2 RELATED WORK

### 2.1 3D-NATIVE GENERATIVE MODELS

Recent advancements in 3D-native generation have been significant. Initial research focused on uncompressed 3D representations, such as Neural Radiance Fields (NeRF) (Jun & Nichol, 2023; Wang et al., 2023), occupancy volumes (Gupta et al., 2023; Cheng et al., 2023; Ntavelis et al., 2023; Zheng et al., 2023; Müller et al., 2023; Cao et al., 2023; Chen et al., 2023a; Yan et al., 2024), and other formats (Liu et al., 2023; Chen et al., 2023b; Yariv et al., 2023; Xu et al., 2024). However, these methods often struggle with small datasets, leading to poor generalization and suboptimal quality. More recent work has transitioned to latent generative models for 3D generation (Zhao et al., 2023; Li et al., 2024; Lan et al., 2024; Hong et al., 2024; Tang et al., 2023; Chen et al., 2024b; Tang et al., 2025). These models use a 3D shape tokenizer to compress raw 3D representations into a more compact latent space suitable for denoising models. This separation allows efficient training of diffusion or flow-based models in the latent space, while a VAE handles reconstruction, enhancing scalability and memory efficiency (Zhang et al., 2024; Wu et al., 2024; Li et al., 2025a; Zhao et al., 2025; Xiang et al., 2024; Ye et al., 2025; Lai et al., 2025; Chen et al., 2025). Conditioning has expanded to include text, single/multi-view images, and other modalities (Zhang et al., 2024; Xiang et al., 2024). Additionally, rectified flow and flow-matching objectives provide alternatives to diffusion training for improved sampling efficiency (Liu et al., 2022; Lipman et al., 2022; Li et al., 2025a).

## 2.2 3D SHAPE TOKENIZERS

Among various 3D representations and auto-encoders explored in latent generative models, two primary types of 3D shape tokenizers are widely adopted.

**Vector set methods.** 3DShape2Vecset (Zhang et al., 2023) first introduced a vector-set latent representation (VecSet) for effective 3D compression. It takes surface point clouds as the input and uses furthest point sampling to obtain query points for the transformer backbone, which learns an unordered vector set as the latent code. This unordered latent representation facilitates training of the denoising model and mesh decoding. Dora (Chen et al., 2024a) further proposes salient edge sampling, which enhances reconstruction fidelity by allowing the network to focus more on sharp edges. While vector set methods have good compression capability and are effective for shape encoding, they suffer from a modality gap between surface point cloud inputs and SDF outputs, which can hinder reconstruction fidelity, especially at high spatial resolution (Zhang et al., 2024; Li et al., 2025a; Zhao et al., 2025; Lai et al., 2025).

**Sparse voxel methods.** Another popular choice is structured latent representation (SLAT), which uses sparse voxels with explicit 3D coordinates and features as the latent code. Trellis (Xiang et al., 2024) first proposes structured latent using a sparse voxel encoder and multi-format (NeRF, Gaussian Splatting, Mesh) decoders to support efficient rectified flow modeling. Sparc3D (Li et al., 2025b) unifies the input and output by directly operating on SDF at the corners of sparse voxels, scaling to higher resolutions and achieving better reconstruction fidelity. By using mixed sparse convolution and local attention mechanisms, sparse voxel tokenizers are more efficient and better at recovering local geometry details. However, they usually require more latent tokens to represent the 3D shape. Additionally, since the number of tokens is variable and is dependent on the 3D shape, an extra stage is needed in the 3D generation process to get the active sparse voxel coordinates, complicating the pipeline and potentially introducing cumulative errors.

Our hybrid voxel set method combines the benefits of both approaches, adopting sparse voxels to enhance reconstruction quality and using vector sets to achieve compressive fixed-length latents.

## 3 METHODOLOGY

Our method involves three main sections. Firstly, we propose algorithms to enhance the efficiency of extracting sparse voxels from raw meshes and converting sparse voxels back to meshes (Section 3.1). Secondly, we discuss the design choices between sparse voxel and vector set methods, and how we combine them into our VoxSet method (Section 3.2). Finally, we describe the image-to-3D flow matching model based on our VoxSet tokenizer (Section 3.3).

### 3.1 SPARSE VOXEL CONVERSION

Similar to Li et al. (2025b), our model is designed to operate on sparse voxels and SDF. For each voxel, we record its 3D coordinates and SDF values at its 8 corners, which are used to perform marching cubes (Lorensen & Cline, 1998) to convert it to triangular meshes. Therefore, we first present efficient algorithms to convert between sparse voxel representation and mesh representation.

**Mesh to Sparse Voxels.** Following previous works (Li et al., 2025b; Zhang et al., 2024), we convert the original mesh into a watertight mesh using an unsigned distance field (UDF) with flood filling. Additionally, we develop efficient CUDA implementations to enhance the efficiency of two crucial steps in the conversion process. Firstly, we use an efficient CUDA binary volume hierarchy (BVH) to calculate the unsigned distance field (UDF). The surface is dilated by the size of one voxel to convert single-layer or non-watertight meshes into watertight meshes. We then implement a CUDA parallel flood filling algorithm through label propagation to get the empty region's mask, and further invert the mask to get the occupied volume and the SDF. Flood filling is typically a single-thread algorithm as it requires sequential traversal of all voxels and maintaining the frontier set of voxels. However, at a resolution of $1024^3$, it takes a long time to finish, and we have to move the volume data to CPU. Instead, we use a label propagation algorithm that is easy to parallelize and can achieve the same result as flood filling. Each voxel is assigned its flattened index as its label, and at each step, we update all voxels' labels to the smallest label of their connected neighbors in parallel. This step is repeated until no labels get updated, and we can use the empty voxel's initial label to retrieve

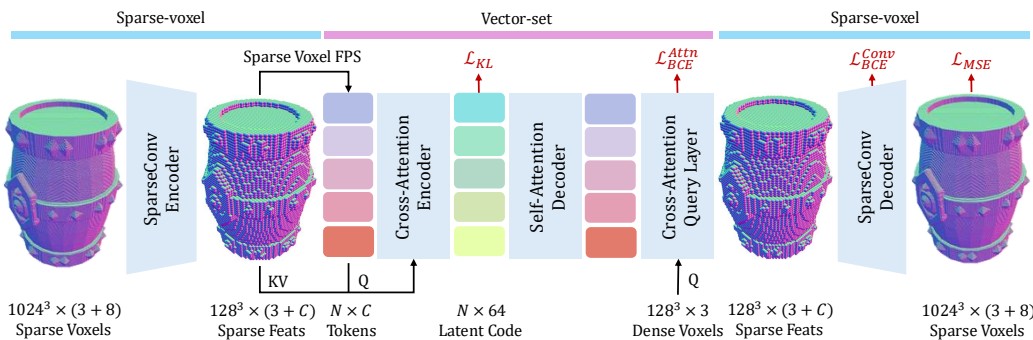

Figure 2: Network Architecture. Our tokenizer is composed of sparse convolution networks outside to extract surface details into sparse features, and a transformer inside to further compress the sparse features into fixed-length latent code.

the empty mask. This efficiently parallels the flood filling and avoids moving data between devices, leading to significant acceleration in processing time.

**Quantization and Storage.** We save the active voxels whose eight corners have different signs and store their coordinates $c \in \mathbb{N}^3$ and the SDF values $\phi \in \mathbb{R}^8$. To reduce storage space, we further pack the coordinates into a single `uint32` number and quantize each corner SDF value into 8 bits, packing all eight values into one `uint64` number. Specifically, since the SDF is used for marching cubes later, we only need the relative value instead of the absolute distance to the nearest surface. We clip the absolute SDF into $[-\sqrt{3}l, \sqrt{3}l]$ where $l$ is the voxel size and quantize it into $[-127, 127]$. An 8-bit signed integer is used to store the value, with one bit for the sign and the remaining seven bits for the magnitude.

**Sparse Voxels to Mesh.** Since the network will generate sparse voxels as described above, we need to convert them back to surface meshes. This requires a sparse variant of the marching cubes algorithm (Lorensen & Cline, 1998). We implement both CPU and CUDA versions of the sparse marching cubes algorithm, which takes the sparse voxels $\{C, \Phi\}$ as input and outputs the vertices and triangles $\{V, F\}$ of the underlying mesh. Since the algorithm operates on each voxel individually, and the imperfect predicted SDF could differ at the same position in different voxels, we also implement a consistency check step to average the values at the same position. All of these steps can operate fully on the GPU to leverage parallel computation, achieving fast conversion speeds from sparse voxels to triangular meshes.

## 3.2 VOXEL SET TOKENIZER

Based on previous works on sparse voxel tokenizers and vector set tokenizers, we observe that the two methods can be complementary. **Sparse voxel tokenizers** are particularly effective at preserving high-quality details, especially at high spatial resolutions. They also unify the input and output format, simplifying data preparation and training. However, these tokenizers require an extra dense volume generation stage due to their use of structured latent representations with explicit 3D coordinates. Additionally, despite a variable length, the average length of the latent code is typically large, resulting in a low compression ratio. **Vector set tokenizers**, in contrast, offer unordered and fixed-length vector set latents that can be directly denoised by the generative model. Furthest point sampling in these tokenizers naturally supports prefix truncation, enabling control over the length of the latent code and facilitating progressive training to accelerate convergence. Despite these advantages, vector set tokenizers face challenges such as misalignment between the output point-occupancy pair and the input, requiring careful mixing of multiple point sampling methods. Moreover, directly querying SDF using a cross-attention layer complicates training and may limit reconstruction quality at high spatial resolutions.

Our key insight is to combine the strengths of both methods. We use sparse voxel convolutions at the outer layers, and in the middle layers where the sparse voxels are downsampled to a lower resolution, we employ the vector set method. By applying furthest point sampling on the sparse voxels and encoding the 3D coordinates implicitly, we compress the them into an unordered vector

set latent code. This hybrid approach leverages the benefits of both techniques: the input and output remain high-resolution sparse voxels, while the latent code is fixed-length and supports truncation for progressive training. As illustrated in Figure 2, our tokenizer is an auto-encoder composed of both sparse convolution layers and attention layers.

**Encoder.** The input to the encoder is the sparse voxels $\{C, \Phi\}$ extracted from watertight meshes. We first apply several residual sparse convolution blocks and sparse downsampling layers (Li et al., 2025b; Xiang et al., 2024) to reduce the resolution and increase the feature channels. These sparse convolution layers can efficiently operate at high resolution (e.g., at $1024^3$, the average number of sparse voxels is about 3 million, while the maximum number can exceed 10 million) to capture local information. Specifically, we downsample the 3D coordinates from $1024^3$ resolution to $128^3$ using sparse average pooling layers. Unlike traditional sparse voxel methods (Li et al., 2025b; He et al., 2025; Chen et al., 2025) which stop here and use a bottleneck block to obtain the latent code, we further compress the sparse voxels using a transformer. We apply furthest point sampling to subsample a fixed-length subset of all sparse voxels based on their coordinates. This subset of voxels serves as the query in the cross-attention encoder, while all the sparse voxels act as the key and value to summarize into our latent code. Similar to vector set methods (Zhang et al., 2023; Chen et al., 2024a), our latent code is an unordered set of vectors, and the length of the latent code can be controlled to balance reconstruction quality and compression ratio. Note that the 3D coordinates are position-embedded into the features, so we do not have explicit coordinates in our latent code.

**Decoder.** Given our voxel set latent code, we apply a series of self-attention layers to decode the latent features. These features serve as the key and value for the subsequent dense occupancy query and sparse feature query at a resolution of $128^3$. Specifically, we first use the positional embedding of each voxel's coordinate to query the occupancy value at this voxel with a cross-attention layer. Only the occupied voxels are kept and further used to query their features with another cross-attention layer. This step retrieves the explicit 3D coordinates from the latent code. During training, we use the ground truth occupancy to prune, ensuring that the output sparse voxels match the ground truth for loss calculation. During inference, we use the predicted occupancy for self-pruning. The output of the cross-attention decoder is sparse voxels at a resolution of $128^3$ with explicit 3D coordinates. We then apply upsampling and sparse convolution layers to gradually increase the resolution back to $1024^3$ to match the input. Similar to Li et al. (2025b), we use self-pruning to remove non-occupied voxels after each upsampling layer to avoid out-of-memory issues at high resolutions. Finally, we apply post-processing to the final mesh to fill small holes produced by inaccurate SDF predictions.

**Training Loss.** Three types of loss functions are used during our tokenizer's training stage. The primary loss is a series of binary cross-entropy losses on the classification of occupancy, $\mathcal{L}_{BCE}$, applied after the cross-attention decoder and each upsampling layer in the sparse convolution blocks. These losses are essential for the model to learn how to perform self-pruning during inference. At the final layer of the decoder, a mean squared error (MSE) loss, $\mathcal{L}_{MSE}$, is applied to supervise the predicted SDF values at the 8 corners of each sparse voxel. This ensures that the sparse marching cubes algorithm can extract valid triangles inside each voxel. Finally, a KL divergence loss, $\mathcal{L}_{KL}$, is used to regularize the latent code. The final loss is a weighted combination of all these losses:

$$\mathcal{L} = \mathcal{L}_{BCE}^{Attn} + \sum_i \mathcal{L}_{BCE}^{Conv_i} + \mathcal{L}_{MSE} + \lambda \mathcal{L}_{KL} \tag{1}$$

This formulation ensures that the model learns effective self-pruning, accurate SDF predictions, and maintains a regularized latent code.

### 3.3 FLOW MODEL

Since our VoxSet tokenizer outputs a fixed-length vector set latent code, we do not need an extra generation stage to produce the sparse structure as required by previous sparse voxel methods (Xiang et al., 2024; Li et al., 2025b; He et al., 2025; Chen et al., 2025). This allows us to take full advantage of vector set methods. Our flow model consists of a stack of attention layers, following similar designs as in (Li et al., 2025a; Zhao et al., 2025; Zhang et al., 2024). We use DINOv2 (Oquab et al., 2023) as the image feature encoder to provide conditioning for the model. The model is trained with an MSE loss to predict the added random noise at each training step. Furthermore, since our latent code can be truncated to balance the trade-off between compression ratio and reconstruction quality, we progressively train the model from fewer to more tokens to accelerate convergence following (Zhang et al., 2024; Zhao et al., 2025; Chen et al., 2024a).

| Flood-filling | | $512^3$ | $1024^3$ |
|---|---|---|---|
| Skimage | CPU | 2.28 | 34.10 |
| Ours | CUDA | **0.16** | **3.72** |

| Marching Cubes | | | $512^3$ | $1024^3$ |
|---|---|---|---|---|
| PyMCubes | CPU | Dense | 2.7538 | 22.2076 |
| Ours | CPU | Sparse | **0.0343** | **0.1796** |
| Diso | CUDA | Dense | 0.0111 | 0.0693 |
| Ours | CUDA | Sparse | **0.0048** | **0.0194** |

Table 1: Comparisons on the time (second) of flood filling and marching cubes under different spatial resolutions.

## 4 EXPERIMENTS

### 4.1 IMPLEMENTATION DETAILS

**Datasets.** We use the Trellis500k subset (Xiang et al., 2024) of the Objaverse-XL dataset (Deitke et al., 2023b;a) (ODC-BY v1.0 license). We further filter the dataset to exclude meshes with no textures or with a low occupancy ratio, resulting in approximately 312K meshes. All meshes are normalized to fit within the $[-0.95, 0.95]^3$ cube, and the sparse voxel conversion is performed at a resolution of $1024^3$. The average sparse voxel extraction time is about 20 seconds, and the average storage size is about 14MB per mesh. Each mesh is rendered from multiple camera viewpoints to serve as the conditional input images. During training, we randomly sample a rendered image at each step, allowing the model to learn pose-invariant generation in the canonical space.

**Tokenizer.** Our tokenizer consists of a sparse convolution encoder, a cross-attention encoder, a cross-attention decoder, and a sparse convolution decoder. The sparse convolution encoder contains 3 residual sparse convolution blocks, each having two sparse convolution layers with $3^3$ kernel size, and a sparse average pooling layer to downsample the sparse voxels. The corresponding resolutions and channels for the blocks are $\{1024, 512, 256, 128\}$ and $\{8, 32, 128, 512\}$. The cross-attention encoder first performs furthest point sampling to sample at most 32768 voxels, and we randomly truncate it to $\{512, 1024, 2048, 4096, 8192, 16384\}$ at each training step. If there are not enough voxels at $128^3$ resolution, we use all available voxels. We concatenate the sinusoidal positional embedding to the voxel features and drop the explicit 3D coordinates. The output features of the cross-attention layer go through a linear layer to compress the channels to 64 and output the mean and standard deviation of our latent code. The cross-attention decoder uses a linear layer to decompress the channels and stacks 8 self-attention layers to decode the features. Two cross-attention layers are used to decode the occupancy and features to convert the vector set to sparse voxels at $128^3$. Finally, the sparse convolution decoder upsamples the sparse voxels back to $1024^3$ resolution and predicts the 8 corner SDF values for each voxel. We use WarpConvNet (Choy & Research, 2025) as the sparse convolution backend. We train the model with the Adam (Kingma & Ba, 2014) optimizer, starting with a learning rate of $10^{-4}$ and decaying to $10^{-5}$. We use 64 A100 GPUs (80G) with a total batch size of 64 for about a week. The weight of the KL loss is set to $10^{-3}$.

**Flow Model.** The flow matching model is a 1.2B DiT (Peebles & Xie, 2022), consisting of 24 blocks. Each attention layer has 1536 channels and 16 heads. We use DINOv2 Giant (Oquab et al., 2023) as the image feature encoder to provide conditioning for the DiT model. The model is trained in multiple stages with an increasing number of latent tokens, following previous works (Zhang et al., 2024; Zhao et al., 2025; Li et al., 2025a; Chen et al., 2024a). We use at most 16384 latent tokens considering the training and inference efficiency. The total training process takes about one week using 256 H100 GPUs. During inference, we use a classifier-free guidance (CFG) scale of 7 and perform 50 steps to generate the latent code.

### 4.2 SPARSE VOXEL CONVERSION

We first evaluate our algorithms for converting between meshes and sparse voxels. The algorithms are benchmarked on a desktop equipped with an AMD Ryzen Threadripper PRO 5975WX 32-Core CPU and an NVIDIA RTX A6000 GPU.

**Parallel Flood-Filling.** We compare the time taken by the flood-filling step using our CUDA implementation and the CPU implementation of skimage (Van der Walt et al., 2014). As shown on

| | #Tokens ↓ | CD ($\times 10^{-5}$) ↓ | F1 ($\times 10^{-2}$) ↑ | ANC ($\times 10^{-2}$) ↓ |
|---|---|---|---|---|
| Dora-1.1 (Chen et al., 2024a) | 4096 | 1.51 | 57.68 | 1.63 |
| Ours | 4096 | **0.98** | **68.81** | **1.05** |
| Dora-1.1 (Chen et al., 2024a) | 8192 | 1.45 | 62.18 | 1.58 |
| Ours | 8192 | **0.57** | **77.73** | **0.71** |
| Ours | 16384 | 0.52 | 79.56 | 0.56 |
| Ours | 32768 | 0.51 | 80.34 | 0.39 |
| SparseCube$^\dagger$ | 34163±13931 | 0.49 | 81.03 | 0.35 |
| SparseFlex (He et al., 2025) | 217970±85766 | 0.50 | 80.72 | 0.36 |

Table 2: Comparison of tokenizer's reconstruction quality. We sample 1M uniform surface points for CD and F1 score. For sparse latent methods, we report the mean and variance of token numbers. $^\dagger$: our implementation of sparse-cube methods similar to Li et al. (2025b) using $128^3$ latent resolution.

the left side of Table 1, our method significantly accelerates the flood-filling process, especially at a resolution of $1024^3$. Note that since the data is originally on the GPU, we don't count the offloading time for the CPU implementation, which would further slow down its speed.

**Sparse Marching Cubes.** We also compare different implementations of marching cubes (Wei et al., 2025; Lorensen & Cline, 1998) to extract the mesh from sparse voxels, as shown on the right side of Table 1. Our algorithms show enhancements under both CPU and CUDA settings by leveraging the 3D sparsity. Note that for dense marching cube implementations, we also don't count the time required to densify the sparse voxels.

## 4.3 RECONSTRUCTION QUALITY

We evaluate the reconstruction quality of the proposed VoxSet tokenizer both quantitatively and qualitatively. The baselines include vector set methods (Chen et al., 2024a) and recent sparse voxel methods (He et al., 2025; Li et al., 2025b).

**Quantitative Comparisons.** To reflect the reconstruction quality on challenging meshes, we curate a test set of high-poly artist-created meshes with complex structures. We use three metrics to evaluate different aspects of the reconstruction quality: Chamfer distance (CD), F1 score (Knapitsch et al., 2017), and absolute normal consistency (ANC) (Li et al., 2025b). For CD and F1 score, we sample 1M uniform surface points from the meshes. All meshes are normalized to $[-0.95, 0.95]^3$ for comparison, and the threshold for the F1 score is set to 0.002 (voxel size at $1024^3$). In Table 2, we show the comparisons between different methods. Our method achieves better reconstruction quality compared to the vector set method (Chen et al., 2024a) using the same number of latent tokens, and is competitive with sparse voxel methods (He et al., 2025) using averagely fewer number of latent tokens. This demonstrates our approach's capability to maintain high fidelity with a more compact representation.

**Qualitative Comparisons.** In Figure 3, we visualize the reconstruction quality and error maps. Consistent with the quantitative results, our method can reconstruct fine details of the surface using fewer tokens. These visual comparisons highlight our method's proficiency in preserving intricate details and achieving smooth surface reconstructions that previous vector set methods find challenging, demonstrating the benefits of combining the two approaches in a unified framework.

## 4.4 GENERATION QUALITY

We further evaluate the image-to-3D generation quality of our flow model trained using our VoxSet tokenizer. In Figure 4, we choose some challenging images in the wild and compare our results against recent methods (Wu et al., 2024; Ye et al., 2025; Zhao et al., 2025). Our method generates more detailed surfaces and usually better follows the input image. This demonstrates that our tokenizer is potential for training a powerful generative model. Its ability to compress complex geometric features into a compact latent space allows for efficient learning and generalization.

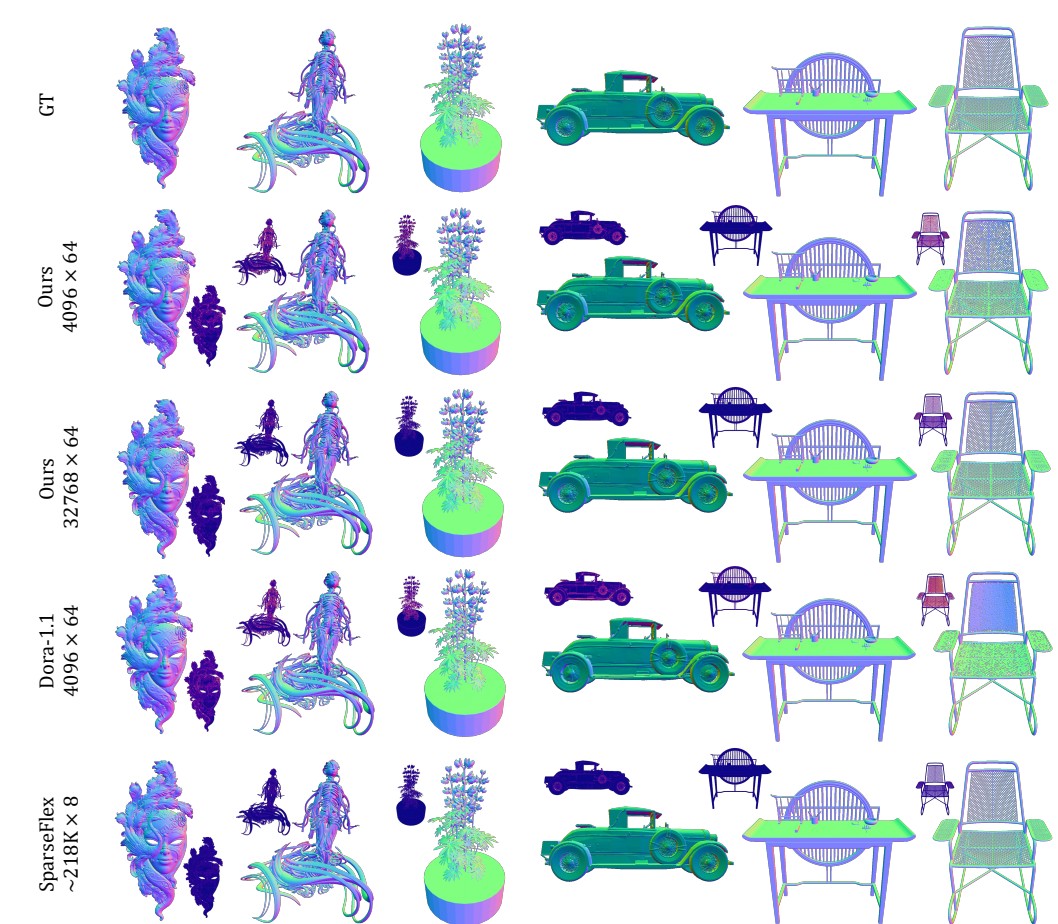

Figure 3: Comparisons on tokenizer's reconstruction quality. We show the error map between rendered normal images for clear comparison.

| Latent Resolution | Latent Size | CD $(\times 10^{-5})\downarrow$ | F1 $(\times 10^{-2})\uparrow$ | ANC $(\times 10^{-2})\downarrow$ |
|---|---|---|---|---|
| $128^3$ | $32768 \times 16$ | 0.54 | 78.65 | 0.65 |
| $128^3$ | $32768 \times 32$ | 0.52 | 79.71 | 0.55 |
| $128^3$ | $32768 \times 64$ | **0.51** | 80.34 | **0.39** |
| $256^3$ | $32768 \times 64$ | **0.51** | **80.41** | 0.44 |

Table 3: Ablation Study on the latent resolution and size.

## 4.5 ABLATION STUDY

**End-to-end Training.** The first design choice is whether to combine the sparse voxel method and vector set method in one framework for end-to-end training, or to perform two-stage training. For example, we could first train a sparse voxel tokenizer (Li et al., 2025b), and then train another vector set tokenizer to further compress the latent code of the first tokenizer. Our early experiments indicate that in the two-stage training approach, the second tokenizer faces difficulty in converging. The latent code of the first tokenizer contains rich-information features, and the reconstruction loss alone is not sufficient for convergence. This is reasonable since we typically need a diffusion model to generate the latent code. Therefore, we adopt end-to-end training in our method to avoid an extra first stage latent code, which proves to be successful in achieving convergence.

**Model Architecture.** We perform experiments to ablate the architecture of our tokenizer, as shown in Table 3. Firstly, we ablate the number of channels in the latent code and found that increasing the

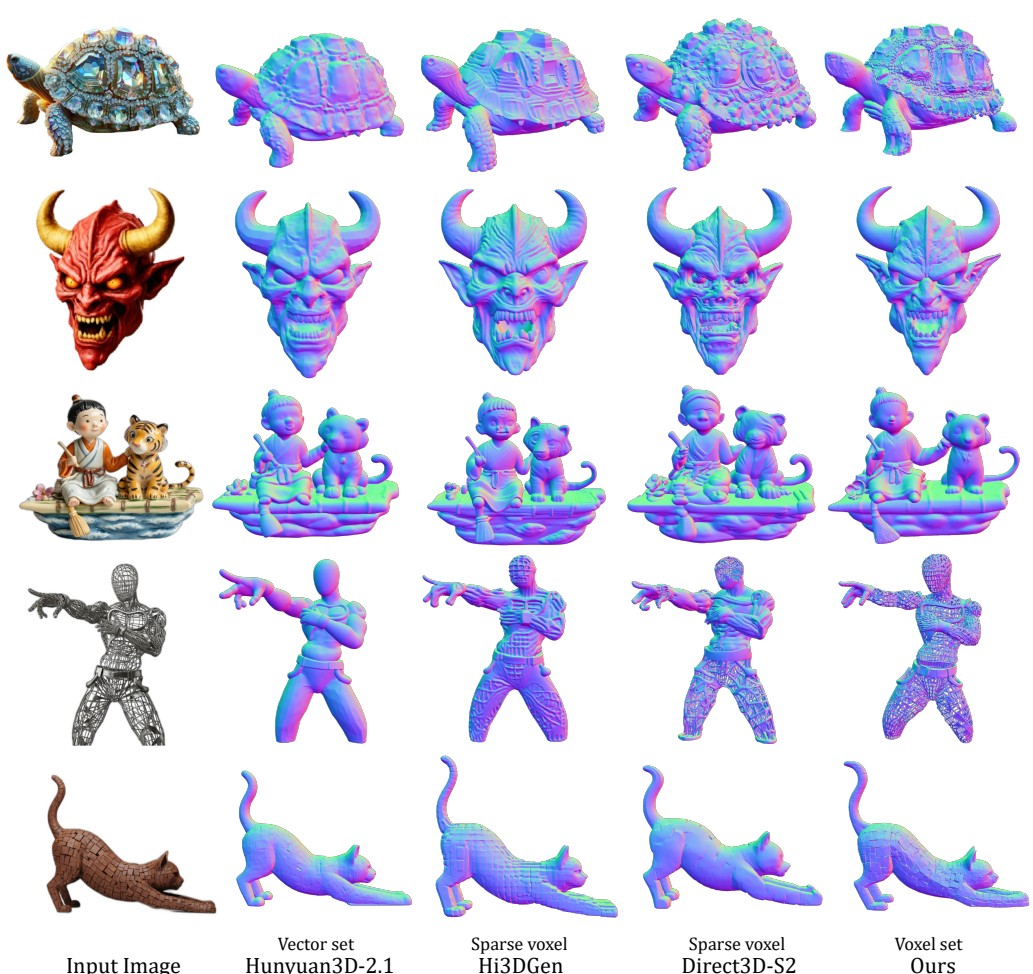

|  | Vector set | Sparse voxel | Sparse voxel | Voxel set |
| Input Image | Hunyuan3D-2.1 | Hi3DGen | Direct3D-S2 | Ours |

Figure 4: Comparisons on image-to-3D generation.

number of channels helps enhance reconstruction quality. We choose 64 channels, consistent with previous works (Chen et al., 2024a; Zhao et al., 2025; Li et al., 2025a). Secondly, we ablate the latent resolution used to sample the voxel set. By removing one block from both the sparse convolution encoder and decoder, we increase the latent resolution to $256^3$. However, the quality enhancement is marginal, so we decide to keep using $128^3$.

### 4.6 LIMITATIONS

There are still limitations to the proposed VoxSet tokenizer. Similar to vector set methods, it requires a relatively longer training time for convergence, and its performance still falls behind state-of-the-art sparse voxel tokenizers (Li et al., 2025b). Our image-to-3D generative model also suffers from robustness issues and may generate unsatisfactory non-watertight shapes due to the noisy dataset.

### 5 CONCLUSION

In this paper, we introduce VoxSet, a hybrid tokenizer that combines sparse voxels with fixed-length vector-set latents to achieve high-fidelity and compact 3D shape auto-encoding. We showcase the potential of VoxSet by training a flow matching model for image-to-3D generation. Experiments demonstrate that VoxSet outperforms vector-set baselines at equivalent token budgets while sharing the straightforward generation pipeline with fixed-length latents, highlighting a promising approach for efficient and scalable 3D generation models.

REPRODUCIBILITY STATEMENT

We described the detailed architecture of the VoxSet tokenizer in Section 3.2 and provided comprehensive details in Section 4.1. Our training dataset is publicly available (Deitke et al., 2023b;a), and we provide the list of testing data in the appendix. To ensure reproducibility and facilitate further research, we commit to open-sourcing the code, model, and necessary scripts in the near future.

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

# A   MORE IMPLEMENTATION DETAILS

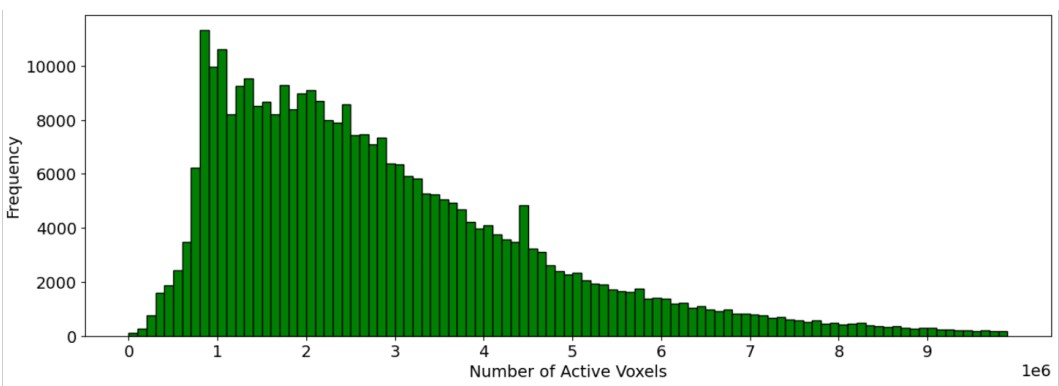

Figure 5: Dataset statistics. We visualize the number of active voxels in our training dataset.

## A.1   DATASET STATISTICS

In Figure 5, we show the statistics of our training dataset. The number of active voxels at a resolution of $1024^3$ has a maximum value of 72 million, with the mean and median values as 2990695 and 2479046 respectively. Due to memory limitation, we only use data with fewer than 10 million voxels. This filters out about 2% of the total data samples.

Our test dataset is manually selected from Sketchfab[1] and consists of high-quality artist-created meshes from after 2022 (Deitke et al., 2023b) to ensure they are not included in the training dataset. We provide the list of our test dataset for the reproduction of our results:

- "Pirate Boat" (https://skfb.ly/pzOWG) by local.yany is licensed under Creative Commons Attribution (http://creativecommons.org/licenses/by/4.0/).

- "Hero Longboat" (https://skfb.ly/pzOWF) by local.yany is licensed under Creative Commons Attribution (http://creativecommons.org/licenses/by/4.0/).

- "Table/Chairs Set – Game Asset" (https://skfb.ly/pyHzq) by AspectStudios is licensed under Creative Commons Attribution (http://creativecommons.org/licenses/by/4.0/).

- "Medieval Furniture" (https://skfb.ly/pzPOt) by Rinku Choudhary is licensed under Free Standard.

- "Venice Mask" (https://skfb.ly/optXA) by DailyArt is licensed under Creative Commons Attribution-NonCommercial (http://creativecommons.org/licenses/by-nc/4.0/).

- "Chinese square table" (https://skfb.ly/pzoMK) by 3Dji is licensed under Creative Commons Attribution (http://creativecommons.org/licenses/by/4.0/).

- "Hussar helmet" (https://skfb.ly/oFGVK) by Virtual Museums of Małopolska is licensed under Creative Commons Attribution (http://creativecommons.org/licenses/by/4.0/).

- "Blue crab" (https://skfb.ly/oKnV8) by C.J..Goldman is licensed under Creative Commons Attribution (http://creativecommons.org/licenses/by/4.0/).

- "Ship of the line, c. 1700 | Linjeskepp, ca 1700" (https://skfb.ly/oT9RF) by SWEDISH NATIONAL MARITIME AND TRANSPORT MUSEUMS is licensed under Creative Commons Attribution (http://creativecommons.org/licenses/by/4.0/).

- "Linden Tree (high-Poly)" (https://skfb.ly/oSKrG) by Sereib is licensed under Creative Commons Attribution (http://creativecommons.org/licenses/by/4.0/).

---

[1] https://sketchfab.com

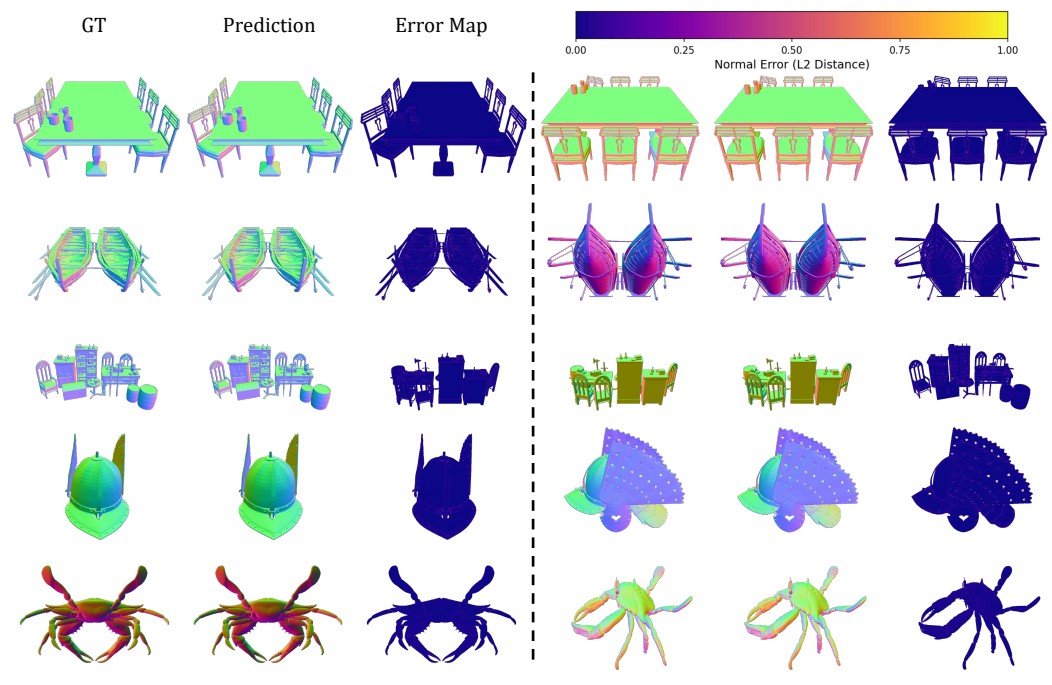

Figure 6: More reconstruction results using our tokenizer.

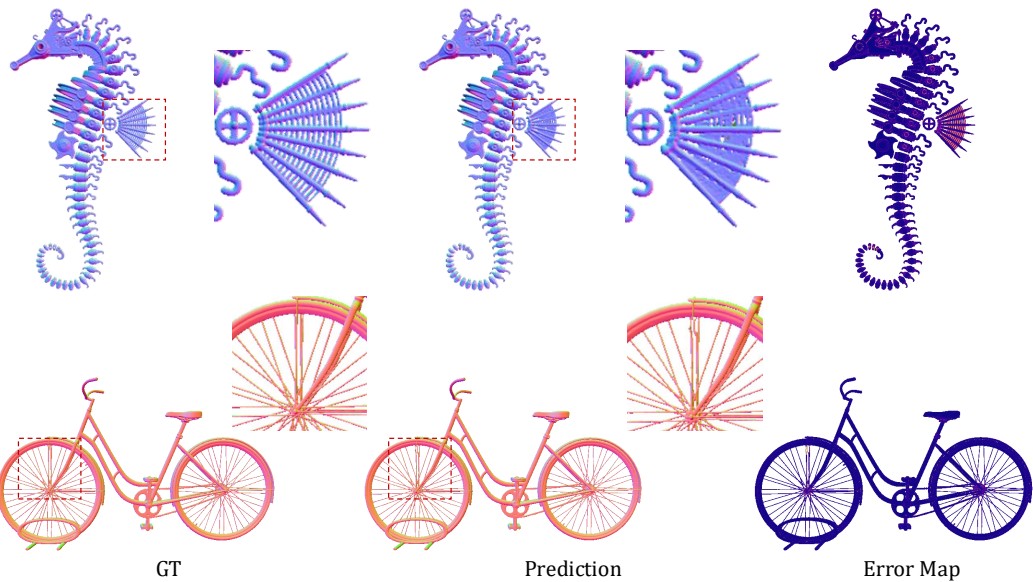

Figure 7: Failure cases of our tokenizer.

## A.2 SPARSE CUBE TOKENIZER

Since most recent sparse voxel methods (Li et al., 2025b) are not open-sourced yet, we attempt to reproduce their results to ensure a fair comparison. We use the same sparse voxel data format as our method, as we observe that introducing vertex deformation and rendering-based optimization leads to more complex and longer data processing times. We follow the architecture described in their paper, using three sparse convolution blocks and a local attention block as the encoder and decoder. Given a $1024^3$ input, the resolution of the latent code is $128^3$, following (Chen et al., 2025). At each upsampling layer, we apply a binary cross-entropy (BCE) loss to supervise the occupancy

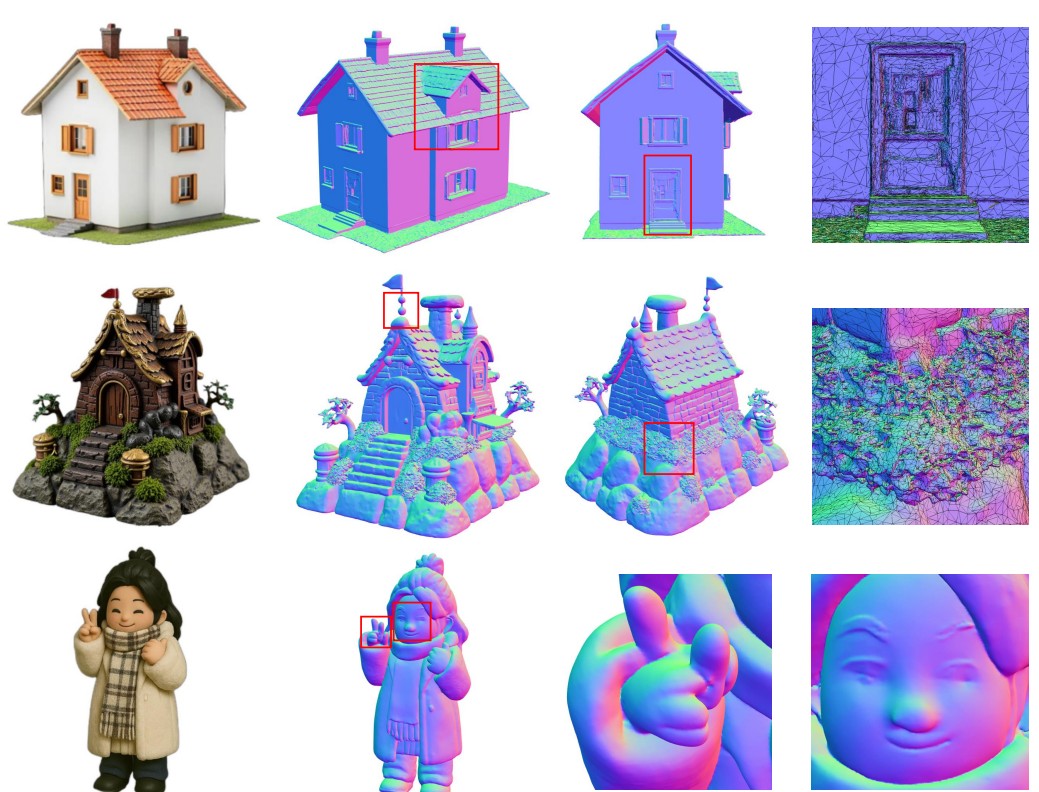

Figure 8: Failure cases of our image-to-3D flow model.

self-pruning. At the last layer, we directly apply mean squared error (MSE) loss to supervise the SDF values instead of separately supervising the magnitude and sign. The model is trained on the same dataset as ours and is denoted as SparseCube[†] in the main paper.

# B    More Results

In Figure 6, we show more reconstruction results of our tokenizer. Our tokenizer can faithfully reconstruct surface details for complex structures, which enhances the performance upper bound of the later generative model. In Figure 7, we also visualize the failure cases of our method. We observe that most failure cases are thin structure which might be hard to capture with the voxel set. In Figure 8, we show some failure cases of our image-to-3D flow model. Typical failure includes messy or broken mesh surface at regions with rich details (such as the grass), and also misalignment with the input image. Since our model is only trained on public datasets which contain few high-quality and high-resolution meshes, we consider this could be enhanced with careful data collection and curation.

