# OpenReview forum: "VoxSet: Sparse Voxel Set Tokenizer for 3D Shape Generation"
_ICLR.cc/2026/Conference — Submitted to ICLR 2026_

### Official Review · Reviewer_3f2H · 2025-10-17

**Soundness:** 3
**Presentation:** 3
**Contribution:** 2
**Rating:** 4
**Confidence:** 4

**Summary:**

This paper introduce VoxSet, a novel tokenizer that combines the strengths of both approaches. The method employs sparse voxels
in the outer layers to capture fine surface details and a vector set bottleneck for high compression. It achieves competitive reconstruction quality compared to sparse voxel tokenizers, while sharing the simpler training and inference pipeline of vector set-based 3D generation models.

**Strengths:**

1. The voxel set tokenizer that combines sparse voxel representation with fixed-length vector set compression, achieving high-fidelity reconstruction within a compact latent code.
2. An efficient implementation.
3. The author conducts experiments on large-scale dataset, demonstrating the superior performance over existing works.
4. The generative quality of the proposed approach is amazing (Figure 4).

**Weaknesses:**

1. The paper directly combines two SOTA 3D shape representations, which is incremental.
2. Although the author provides quantitative comparisons on reconstruction, such comparison on generation is missing. The author compares Hunyuan3D et al qualitatively but not quantitatively.
3. The performance still falls behind state-of-the-art sparse voxel tokenizers, it shows that the proposed design stills sacrifices the quality.
4. It would be better is the author provides more experiments like text-to-shape.

**Questions:**

See weakness

---

> ### Author Response · Authors · 2025-11-18
>
> Thank you for your valuable time and insightful comments! We have tried to address your concerns in the updated manuscript and our rebuttal text:
>
> **Q1: Contributions are incremental.**
>
> We argue that integrating sparse voxels with vector sets is non-trivial. Our experiments show that this hybrid preserves the strengths of both while mitigating their weaknesses. We believe these results offer valuable insights for the field.
>
> **Q2: Lack of quantitative comparisons and text-to-3D results for generation.**
>
> We emphasize that our work focuses on the tokenizer rather than the flow model. In line with recent 3D tokenizer studies (Dora, Sparc3D), we use the flow model to demonstrate the tokenizer’s capability, not to compete for state-of-the-art generative performance. Quantitative comparison of generation quality remains an open problem without agreed-upon metrics, and many leading models are commercial and closed-source, complicating fair evaluations. In practice, text-to-3D is typically approached via text-to-image followed by image-to-3D, rather than trained as a separate end-to-end model.
>
> **Q3: Performance gap with sparse voxel tokenizers.**
>
> We acknowledge that VoxSet shows a slight performance gap relative to state-of-the-art sparse-voxel tokenizers. However, its primary strength and our core motivation is the fixed latent-token budget, which is critical for an end-to-end latent generation framework. In contrast, SparseCube produces a variable number of tokens, necessitating an additional sparse-structure generation stage, as in Trellis and Sparc3D.

---

> ### Author Response · Authors · 2025-11-28
>
> We sincerely appreciate your great efforts in reviewing this paper. Your constructive advice and valuable comments really help improve our paper. Considering the approaching deadline, please, let us know if you have follow-up concerns. We sincerely hope you can consider our reply in your assessment, and we can further address unclear explanations and remaining concerns if any.
>
> Once more, we are appreciated for the time and effort you've dedicated to our paper.

---

### Official Review · Reviewer_wZ25 · 2025-10-25

**Soundness:** 2
**Presentation:** 2
**Contribution:** 3
**Rating:** 4
**Confidence:** 3

**Summary:**

This paper proposes VoxSet, a novel 3D tokenizer that combines the strengths of sparse voxel and vector set representations, achieving both high-quality and compact format. Experiments demonstrate reconstruction and image-to-3D generation capabilities of VoxSet. Moreover, this paper proposes an algorithm optimization for sparse voxel conversion stages and shows its efficiency.

**Strengths:**

The motivation of this paper is clear, and the proposed VoxSet effectively combines the advantages of sparse voxel and vector set representations.
The experimental results demonstrate the effectiveness of VoxSet in 3D shape reconstruction and image-to-3D generation tasks, showing its potential for practical applications.

**Weaknesses:**

The paper lacks detailed explanations of the proposed methods, for example, the comparison between different tokenizers and the image-to-3D generation flow. This makes it difficult to fully understand the contributions and novelty of the work.
The tokenizer requires a long training time, which is not friendly for hyperparameter tuning in practice.
The performance improvement compared to voxel-based methods is not very significant, requires further justification of the efficiency of the proposed method.

**Questions:**

1. The methods in Section 3.1 are several straightforward algorithm optimizations, it's better to present a pseudo code (maybe in the Appendix) to illustrate the algorithm. Additionally, provide more information about previous works and consider analyzing why they did not utilize similar optimizations to demonstrate the novelty.
2. It's better to provide comparisons in Figure 2 with the sparse voxel tokenizer and vector set tokenizer, to clarify their differences between the proposed sparse voxel tokenizer.
3. Section 3.3 is a bit brief, which doesn't clarify the image-to-3D generation process. For example, what is the role of the tokenizer in the image-to-3D generation, and where is its input from and output to?
4. As shown in Table 2, VoxSet has a similar performance and token count to the sparsecube method. How about the quantitative efficiency comparison between these two methods?

---

> ### Author Response · Authors · 2025-11-18
>
> Thank you for your valuable time and insightful comments! We have tried to address your concerns in the updated manuscript and our rebuttal text:
>
> **Q1: More details on the image-to-3D generation process.**
>
> Thank you for the suggestion! Our model follows a standard tokenizer & latent-flow (diffusion) framework, analogous to text-to-image systems. The tokenizer compresses a 3D shape (e.g., a triangle mesh) into a compact latent code and reconstructs it as a voxel-set representation. We then train a latent flow model conditioned on a single-view input image to predict the corresponding 3D latent, which is decoded back to the 3D shape.
>
> **Q2: The tokenizer requires a long training time, and the performance improvement is not very significant.**
>
> The tokenizer’s role is to produce a compact latent code for the downstream latent-flow generator. Compared with voxel-based approaches that use a variable number of latent tokens, our method removes the sparse-structure generation stage, which is usually an additional latent-flow model that is far more expensive than a tokenizer. As a result, we need only one latent-flow model for image-to-3D generation, whereas sparse-voxel methods (e.g., Trellis, Sparc3D) require two: (1) image → sparse structure, and (2) image + sparse structure → 3D shape.
>
> **Q3: More details on the optimized algorithms in Section 3.1.**
>
> Thank you for the suggestion. These algorithms are deterministic and straightforward, but there are currently no open-source implementations. Because flood filling and marching cubes are typically treated as post-processing after model inference, prior work seldom details them. We will release our implementations as a Python library for easy integration in future research.
>
> **Q4: Compare the structure of sparse voxel tokenizer in Fig 2.**
>
> Good idea! We have revised the manuscript and updated the figure to more clearly highlight the differences. In essence, compared with a pure sparse-voxel tokenizer, our method adds only an intermediate vector-set bottleneck and the transition layers.
>
> **Q6: Quantitative efficiency between VoxSet and SparseCube.**
>
> First, we clarify that SparseCube is VoxSet without the intermediate vector-set bottleneck (see Fig. 2). Tab. 2 provides quantitative results, and we acknowledge that the bottleneck in VoxSet causes a small performance drop relative to SparseCube. The key strength and motivation of VoxSet is its fixed latent-token budget, which is critical for an end-to-end latent generation framework. By contrast, SparseCube produces a variable number of tokens, necessitating a separate sparse-structure generation stage, as in Trellis and Sparc3D.

---

> > ### Comment · Reviewer_wZ25 · 2025-11-27
> >
> > Thank you for taking the time to respond to my comments. The quantitative efficiency that I mentioned is the time efficiency (latency comparison using VoxSet and SparseCube). Could you please provide more information about this?

---

> > > ### Author Response · Authors · 2025-11-28
> > >
> > > Thank you for the reply, and apologies for the misunderstanding!
> > > Our method is essentially SparseCube with an additional VecSet bottleneck. The VecSet component adds only about 10% overhead during inference:
> > > | Component          | Time (ms) |
> > > | ------------------ | --------- |
> > > | SparseConv Encoder | 554       |
> > > | VecSet Encoder     | 64        |
> > > | VecSet Decoder     | 164       |
> > > | SparseConv Decoder | 1665     |
> > > Overall, the full encoding–decoding pipeline takes roughly 2.5 seconds, which is sufficiently fast compared to downstream post-processing steps such as mesh decimation.

---

> > > > ### Comment · Reviewer_wZ25 · 2025-11-28
> > > >
> > > > Thank you for the additional information. But my focus is more about the quantitative analysis of the statement "The key strength and motivation of VoxSet is its fixed latent-token budget, which is critical for an end-to-end latent generation framework." If VoxSet introduces extra time overhead to SparseCude, why is it more efficient in the end-to-end latent generation framework? Could you please provide quantitative results to justify this?

---

> ### Author Response · Authors · 2025-11-28
>
> Thank you for the clarification. To avoid confusion, the “efficiency” we refer to here concerns the **diffusion-based generation stage**, not the previously discussed tokenizer stage.
>
> - Our pipeline requires **only a single image-to-latent diffusion model**, which follows the same lightweight 1D DiT architecture used in prior VecSet models.
> - In contrast, sparse-voxel methods typically require **two separate generative models**:  (1) an image-to-sparse-structure model, and (2) a sparse-structure–conditioned diffusion model that predicts the final sparse latent. The second model is substantially more complex. Its architecture departs from the simple 1D DiT design and instead relies on **3D sparse windowed attention**, which is significantly harder to train and runs slower at higher resolutions.
>
> As a result, our generation stage (inherited from VecSet methods) is generally more efficient than the two-stage pipelines used by sparse-voxel approaches. However, providing a fair, quantitative comparison is difficult due to the differences in network architectures and diffusion steps (quality-speed trade-off). Moreover, many sparse-voxel systems are closed-source, limiting direct evaluation.
>
> To offer a rough sense of runtime:
> - **Our model** typically takes **~20s for 30-step diffusion inference**, **~3s for tokenizer decoding**, and **20–60s for post-processing** (post-processing time varies depending on mesh complexity and decimation algorithm).
> - **Hitem3D** (a commercial system using the Sparc3D tokenizer, similar to SparseCube in paper) generally takes **around 5 minutes** end-to-end via its web API. This includes model inference, post-processing, and network latency, so the numbers are not directly comparable.

---

### Official Review · Reviewer_Gn5s · 2025-10-30

**Soundness:** 3
**Presentation:** 2
**Contribution:** 2
**Rating:** 2
**Confidence:** 5

**Summary:**

The paper combines the Shape2VectorSet-like Perceiver IO tokenization paradigm with sparse voxel encoder-decoder, which is stated to have enjoy the strenghs from both perspectives: (1) The latent compactness of S2VS; (2) The ability of representing details via Sparse Voxel Convolution. It represents reconstruction comparison with others to demonstrate the superb geometry compression ability. Moreover, it shows an Image-to-3D experiment to show the feasibility of generating such compact voxel latent tokens.

**Strengths:**

**The core argument is fair**: TRELLIS-like sparse voxel does have the need to generate sparse voxel locations first and then generate the associated latents. To tackle this, the author put a Perceiver IO-like structure within the bottoleneck of sparse voxel encoder-decoder network, enabling a fixed length compact representation. The sparse voxel encoder-decoder in-return will improve the reconstruction details compared to pure S2VS-like methods.

**Weaknesses:**

1. The **presentation** in the paper:

    a. Section 3.1, it uses few long paragraphs to mention the **implememtation details**. While it is appreciated to show details of implementation, it is strongly recommended to move the technical details to supplememntary.

    b. The more important question is that if those implementations really **have something new or just engineering**? As a graphics people, I do think algorithm efficiency is one of the keys of graphics system application, but based on my evaluation, the paper is not inventing something new here, which makes the point stronger that it should move those contents to supplementary/appendix section.
 --------
&nbsp;

2. *The two-edge sword*:
     a. It is a fact that we do need two stages of generation in TRELLIS-like sparse voxel generation task. However, it is proved that we can decode to various representation based on a unifed structral latent. On the other hand, this paper only shows the geometry generation instead of incooparating material generation etc.

     b. **More importantly**:  Although it is an important beneficial to have a more compact 3D representation, it should also be stated that is essentially beneficial for generation of 3D based on that representation. That is why it is recommended to measure rFID as well as FID in 2D image tokenization and generation task. So when we look at Figure 4., which is the sole 3D generation comparison in the paper, we can notice an *very important issue* of Image-to-3D demo of Voxel Set, **Alignment**:

          i . **Third row tiger-child-statue**: the human head and tiger orientation alignment with the input image of Voxel Set is not only incorrect but also obviously worse than Huyuan3D and Hi3DGen.
         ii. **The last row**: Some big cracks on the cat body shows on Huyuan3D-2.1 but not on Voxel Set.

     In all, it makes very hard to clearly demonstrate it is a better representation for generation. After all, **alignment** with input images is the most important feature we want to see for an Image-to-3D task.
 --------

&nbsp;

3. Lack of Spark3D/Hitem3D comparison in reconstruction and generation: I think it is an important baseline both in recosntruction and generation and it is being mentioned in introduction but I am not sure why it is not compared somehow.

**Questions:**

1. As raised in the weakness session, we need more baseline comparisons. Especially, Spark3D/Hitem3D.

2.  It is crucial to dig further on the generation ability based on Voxel Set. I recommend you design more experiments to find out why the alignemt is the issue.

---

> ### Author Response · Authors · 2025-11-18
>
> Thank you for your valuable time and insightful comments! We have tried to address your concerns in the updated manuscript and our rebuttal text:
>
> **Q1: More comparisons with Sparc3D/Hitem3D.**
>
> First, because Sparc3D is not open-sourced, we compare reconstruction quality against our partial reproduction (SparseCube). Second, Hitem3D is a commercial system trained on private datasets, making fair comparison on public data difficult. **We want to emphasize that our paper focuses on the tokenizer rather than the latent generative model.** Following Dora and Sparc3D, the generation results are presented as evidence of the approach’s potential.
>
> **Q2: Lack of exploration on the alignment in image-to-3D.**
>
> To reiterate, we are not competing for the best generative model. The observed alignment imperfections likely stem from training dataset characteristics and model scale, which are more fundamental factors beyond this work’s scope. Even top commercial systems exhibit misalignment frequently. Our study focuses on tokenizer design, not on solving generative alignment.
>
> **Q3: The method can only decode to geometry.**
>
> Most prior work treats geometry and material generation separately. While Trellis can decode to GS or NeRF for appearance, its material fidelity lags behind dedicated multi-view image generation models. Moreover, material generation is orthogonal to the two-stage generation limitation of sparse-voxel methods. For example, Sparc3D still uses a two-stage latent flow pipeline for geometry alone.

---

> ### Author Response · Authors · 2025-11-28
>
> We sincerely appreciate your great efforts in reviewing this paper. Your constructive advice and valuable comments really help improve our paper. Considering the approaching deadline, please, let us know if you have follow-up concerns. We sincerely hope you can consider our reply in your assessment, and we can further address unclear explanations and remaining concerns if any.
>
> Once more, we are appreciated for the time and effort you've dedicated to our paper.

---

### Official Review · Reviewer_kav5 · 2025-11-05

**Soundness:** 3
**Presentation:** 3
**Contribution:** 2
**Rating:** 4
**Confidence:** 4

**Summary:**

The paper introduces VoxSet, a Sparse Voxel Set Tokenizer for 3D shape generation, designed to combine the strengths of two dominant 3D tokenization methods: sparse voxels and vector sets. This is a very interesting work.

**Strengths:**

The core strength is the novel hybrid design that integrates sparse voxels for high-resolution detail capture with a fixed-length vector set bottleneck for efficient compression and a simpler generation pipeline.

- The method achieves competitive reconstruction quality, outperforming vector set methods (like Dora-1.1) at the same token count and remaining competitive with state-of-the-art sparse voxel methods (like SparseFlex and SparseCube) with fewer average tokens.

- The paper is well-structured and easy to follow.

**Weaknesses:**

- While competitive, the reconstruction quality (F1-score) of VoxSet still slightly lags behind the current state-of-the-art sparse voxel tokenizers like SparseCube and SparseFlex, especially at their average token counts.

- The comparison in Table 2 shows that your method performs well with 4096-32768 fixed tokens, while SparseCube and SparseFlex use variable lengths with much larger averages ($\approx 34K$ and $\approx 218K$). While impressive for compression, could you provide results for VoxSet with a higher, but still fixed, token count (e.g., $65536 \times 64$) to see if it can fully close the minor quality gap with SparseCube's $34K$ average, or if the limitation is inherent to the vector-set bottleneck? And what about the memory consumption?

- Your method uses sparse convolution layers (voxel-based) at the outer layers and a transformer (vector-set based) in the middle. Have you performed an ablation to quantify the contribution of each component to the final reconstruction quality? For example, comparing a parse-voxel-only version at resolution (before the transformer) to the full VoxSet model.

- What about the training time and inference time for the voxset？

- For the proposed accelerated voxel-to-mesh algorithm, Figure 1 claims super improvements in the time efficiency. What about the final mesh quality? Is it comparable?

- For the visual comparison in figure 3, could you demonstrate the performance under the same number of tokens?

- I notice that the proposed method has the ability to reconstruct the thin structure in figure 3. What about the noisy input?

- You mention downsampling the $1024^3$ input to $128^3$ using sparse average pooling. Could you elaborate on how the sparse average pooling is implemented in detail? Specifically, how is the average calculated when the surrounding voxels might be nonexistent (sparse)?

- There are similar work (Geometry Distributions) published at ICCV 2025; some discussion and comparison on final mesh quality should be evaluated.

**Questions:**

See weaknesses

---

> ### Author Response · Authors · 2025-11-18
>
> Thank you for your valuable time and insightful comments! We have tried to address your concerns in the updated manuscript and our rebuttal text:
>
> **Q1: Performance is still worse than sparse-voxel methods.**
>
> We acknowledge that even with more latent tokens, our performance remains slightly behind the best sparse-voxel methods. Given our minimum voxel resolution of 128^3 (averaging ~37K sparse voxels on Objaverse), increasing the token budget beyond 32768 offers no further gains and only raises inference cost. We believe this small trade-off is justified by using a fixed latent-token budget, which eliminates the need to train an additional sparse-structure generation model.
>
> **Q2: Ablation with a sparse-voxel only version.**
>
> To clarify, the SparseCube baseline in our experiments is an ablation of our VoxSet model without the vector-set bottleneck. Because there is no open-source sparse-voxel tokenizer, we implemented a Sparc3D-style variant, omitting details that are unclear in the paper (e.g., vertex offsets and rendering-based optimization). In line with Q1, the vector-set bottleneck introduces a small performance degradation.
>
> **Q3: Training time and inference time for the VoxSet.**
>
> We trained on 64 A100 GPUs for about one week. As with other sparse-voxel methods, training is memory-intensive, so we use a per-GPU batch size of 1. At inference, the encoder and gradients are not required, reducing memory demand. The tokenizer decoding inference only takes about 5 seconds, though subsequent post-processing—especially triangle decimation—can be more time-consuming.
>
> **Q4: For the accelerated voxel-to-mesh algorithm, is the performance comparable?**
>
> Yes. Both flood-filling and sparse marching cubes are deterministic. Our speedup comes from a parallel CUDA implementation, which preserves correctness.
>
> **Q5: In Fig. 3, could you demonstrate the performance under the same number of tokens?**
>
> Fig. 3 reports token counts and compares our method with Dora at matched token count. For sparse-voxel methods like SparseFlex, the token count is fixed and cannot be adjusted.
>
> **Q6: Noisy input reconstruction.**
> Our input is a triangle mesh (converted to an SDF), so noisy input would typically mean non-watertight geometry, holes, or broken triangles—cases that are not commonly explored in prior 3D tokenizer work. The primary goal of our tokenizer is to produce a compact latent code for the downstream flow model.
>
> **Q7: Sparse average pooling implementation details.**
>
> We follow Trellis, aggregating only existing neighbors via averaging using `torch.scatter_reduce`. For example, if a 2×2 grid contains only one occupied voxel, its feature is passed unchanged to the next hierarchy.
>
> **Q8: Discussing Geometry Distributions.**
>
> Thank you for the suggestion! We have revised the manuscript to broaden the related-work discussion. Geometry Distributions explores a new 3D representation, whereas our focus is still on typical VAE-based 3D tokenizers.

---

> ### Author Response · Authors · 2025-11-28
>
> We sincerely appreciate your great efforts in reviewing this paper. Your constructive advice and valuable comments really help improve our paper. Considering the approaching deadline, please, let us know if you have follow-up concerns. We sincerely hope you can consider our reply in your assessment, and we can further address unclear explanations and remaining concerns if any.
>
> Once more, we are appreciated for the time and effort you've dedicated to our paper.

---

### Official Review · Reviewer_LuPr · 2025-11-11

**Soundness:** 4
**Presentation:** 4
**Contribution:** 4
**Rating:** 8
**Confidence:** 3

**Summary:**

This paper introduces VoxSet, a hybrid 3D shape tokenizer that combines the advantages of sparse voxel and vector set representations for efficient 3D generative modeling. The authors argue that existing methods face a trade-off between fidelity (sparse voxels) and compactness (vector sets). VoxSet integrates sparse convolution layers to capture fine local geometry and employs a transformer-based vector-set bottleneck for fixed-length latent compression. The method maintains the reconstruction quality of sparse voxel tokenizers while simplifying the generation pipeline to a single stage. Extensive experiments on the Trellis500k dataset show competitive or superior reconstruction results to vector-set baselines and close performance to state-of-the-art sparse voxel tokenizers, with quantitative and qualitative evaluations. The authors also provide CUDA-based acceleration for voxel-mesh conversion.

**Strengths:**

1. Well-motivated hybrid design effectively combining sparse voxel and vector-set representations.
2. GPU-accelerated voxel–mesh conversion (CUDA flood fill and sparse marching cubes) significantly improves preprocessing and inference speed.
3. Simplified single-stage generative pipeline by removing variable-length latent encoding.
4. Comprehensive experiments (quantitative, qualitative, and ablation) showing strong reconstruction with fewer latent tokens.
5. Potential applicability to future 3D diffusion and flow-based generative models.

**Weaknesses:**

1. Limited conceptual novelty — mainly a pragmatic hybridization rather than a new theoretical idea.
2. Overemphasis on implementation details over architectural or theoretical insights.
3. Missing scalability and robustness analysis, especially for noisy or imperfect meshes.
4. Shallow flow model evaluation and lack of strong baseline comparisons.
5. Insufficient analysis of latent space behavior and trade-offs between token efficiency, fidelity, and complexity.

**Questions:**

1. How sensitive is VoxSet to voxel resolution and latent token count during training and inference? Does performance degrade gracefully with fewer tokens?
2. Would the hybrid approach still be beneficial if the generative prior directly operated on sparse voxel structures (as in Trellis or Sparc3D)?

---

> ### Author Response · Authors · 2025-11-18
>
> Thank you for your valuable time and insightful comments! We have tried to address your concerns in the updated manuscript and our rebuttal text:
>
> **Q1: Performance sensitivity to voxel resolution and latent token count in training and inference.**
>
> In summary, the latent token count remains a key driver of performance.
>
> First, the token count scales with the minimum voxel resolution. For example, in Objaverse, 128^3 resolution averages ~37K occupied voxels, so we cap the latent tokens at 32768. Table 3 analyzes the impact of voxel resolution, and Fig. 1 examines the effect of token count. As in prior vector-set methods, we train with randomly sampled token numbers to support flexible truncated inference and observe performance degradation under truncation, but our method still outperforms previous vector-set baselines with 4096 tokens. This suggests that the gains in sparse-voxel approaches stem from both the voxel representation and the increased latent token count.
>
> **Q2: Would the hybrid approach still be beneficial if the generative prior directly operated on sparse voxel structures?**
>
> Our hybrid approach is designed specifically to avoid denoising directly on sparse voxels. There are two main drawbacks to operating on sparse voxels: (1) It requires a pre-generated sparse structure before denoising, adding another stage that can introduce cumulative error. (2) Methods like Trellis and Sparc3D run at 64^3, while recent work such as Ultra3D diffuses at 128^3, which has 4–5× more occupied voxels than 64^3 and is substantially slower to train. To our knowledge, there is still no open-source reproduction of sparse-voxel diffusion at 128^3. Our approach offers a practical path to higher resolution while controlling the number of latent tokens.
>
> **Q3: Shallow flow model evaluation and lack of strong baseline comparisons.**
>
> We want to emphasize more on the tokenizer instead of the flow model. Also, since recent strong flow models are not open-sourced and likely trained on different datasets, we found it hard to do fair comparisons.
>
> **Q4: Missing scalability and robustness analysis.**
>
> Thank you for the suggestion! We present tokenizer failure cases in Fig. 7 and now have added failure cases for the generation model, clarifying the method’s limitations.

---

> ### Author Response · Authors · 2025-11-28
>
> We sincerely appreciate your great efforts in reviewing this paper. Your constructive advice and valuable comments really help improve our paper. Considering the approaching deadline, please, let us know if you have follow-up concerns. We sincerely hope you can consider our reply in your assessment, and we can further address unclear explanations and remaining concerns if any.
>
> Once more, we are appreciated for the time and effort you've dedicated to our paper.

---

### Author Response · Authors · 2025-11-30
**Summary of rebuttal and clarification on scope**

We are writing to provide a summary of our rebuttal and clarify key aspects of our submission. We hope this summary assists you in your final assessment.

While Reviewer LuPr (Rating: 8) correctly identified our core contribution, **a novel 3D tokenizer that combines high-fidelity reconstruction quality with a simplified single-stage generation pipeline**, other reviewers expressed concerns stemming from a misunderstanding of the paper's scope and the inherent limitations of existing sparse-voxel baselines.

We wish to clarify two critical points:

**Tokenizer Design vs. Generation Capability**: Reviewers Gn5s and 3f2H penalized the paper based on the semantic alignment or quality of the Image-to-3D generation results.

Our primary contribution is the 3D Tokenizer itself, focusing on latent compactness and reconstruction quality. The downstream generative model is presented solely to demonstrate that our tokenizer is compatible with standard diffusion transformers (DiT). Issues with semantic alignment or consistency in generation are more generally attributed to the scale of training data and the capacity of the generative model, and is an open problem with no true solution. We believe that criticizing a tokenizer methodology based on the alignment performance, and expecting results on par with commercial models is out of scope for this contribution. Reviewer Gn5s (Confidence: 5) also asked for a comparison with Spark3D (likely refering to Sparc3D, "c" for "cube"). While this method is closed-source, we have partially reproduced it denoted as "SparseCube" in our paper as an important baseline.

**Single-Stage vs. Two-Stage Generation**: Reviewers kav5 and wZ25 questioned the preference for VoxSet given its slightly lower reconstruction quality compared to pure sparse-voxel methods (e.g., Sparc3D), and questioned the efficiency gains.

Pure sparse-voxel methods rely on variable-length tokens. This necessitates a complex, two-stage generation process: (1) generating the sparse structure (active voxels), followed by (2) generating the sparse latents conditioned on that structure.
In contrast, VoxSet employs a fixed-length vector-set bottleneck. This allows us to bypass the difficult "structure generation" stage entirely, enabling a single-stage pipeline using standard 1D DiT architectures and sparse-voxels. Reviewer wZ25 appears to have misunderstood our efficiency claims by comparing only tokenizer inference times; however, our claimed efficiency gain is at the system level. By eliminating the need for a separate, computationally expensive structure-generation model, the overall generation pipeline becomes significantly simpler and more efficient to train and deploy.

**In conclusion**, we want to emphasize that VoxSet is the first to explore the integration of sparse voxel convolution with vector-set latent representations, effectively bridging the gap between high-fidelity reconstruction and generative convenience. We believe this work provides critical insights into the important problem of 3D tokenization and representation.

---

### Meta-Review · Area_Chair_kmhk · 2026-01-06

**Summary:**

VoxSet is a hybrid 3D shape tokenizer that combines sparse-voxel encoding/decoding with a fixed-length transformer-based vector-set bottleneck. The paper received initial scores of 4/2/4/8/4 and remained unchanged before the reset.

Conceptual novelty is viewed as limited and largely an incremental/pragmatic hybridization with heavy emphasis on engineering/implementation over new architectural or theoretical insight (Reviewers LuPr, Gn5s, 3f2H). Multiple reviewers request stronger and more complete baselines and generation evaluation (e.g., missing Spark3D/HiTem3D and other strong comparisons; shallow flow/image-to-3D evaluation; qualitative-only or unclear generation evidence) (Reviewers LuPr, Gn5s, wZ25, 3f2H). Besides, the paper is criticized for insufficient analysis of scalability/robustness, limited latent-space and token–fidelity trade-off study, and incomplete reporting of practical costs (Reviewers LuPr, kav5, wZ25, 3f2H). The rebuttal may not fully address the critical issues from several reviewers, including more comparisons (Reviewers kav5, Gn5s, wZ25, 3f2H), better representation for generation (Reviewer Gn5s), incremental novelty (Reviewers LuPr, 3f2H) etc.

After a careful assessment of the submission, reviews, response, and discussion, the AC recommends rejection. The authors are encouraged to revise and refine the manuscript in accordance with the reviewers’ feedback for a future submission.

**Reviewer Concerns:**

Conceptual novelty is viewed as limited and largely an incremental/pragmatic hybridization with heavy emphasis on engineering/implementation over new architectural or theoretical insight (Reviewers LuPr, Gn5s, 3f2H). Multiple reviewers request stronger and more complete baselines and generation evaluation (e.g., missing Spark3D/HiTem3D and other strong comparisons; shallow flow/image-to-3D evaluation; qualitative-only or unclear generation evidence) (Reviewers LuPr, Gn5s, wZ25, 3f2H). Besides, the paper is criticized for insufficient analysis of scalability/robustness, limited latent-space and token–fidelity trade-off study, and incomplete reporting of practical costs (Reviewers LuPr, kav5, wZ25, 3f2H). The rebuttal may not fully address the critical issues from several reviewers, including more comparisons (Reviewers kav5, Gn5s, wZ25, 3f2H), better representation for generation (Reviewer Gn5s), incremental novelty (Reviewers LuPr, 3f2H) etc.

**Reviewer Scores:**

The manuscript received initial review scores of 4/2/4/8/4. After the rebuttal/discussion and before the reset, the score remained unchanged.

Since several concerns raised by the reviewers may remain unresolved after the rebuttal (see 'Reviewer Concerns'), I would approximate 4/2/4/6/4 as the final score.

---

### Decision · Program_Chairs · 2026-01-26

Reject